# Development of a Colorimetric Tool for SARS-CoV-2 and Other Respiratory Viruses Detection Using Sialic Acid Fabricated Gold Nanoparticles

**DOI:** 10.3390/pharmaceutics13040502

**Published:** 2021-04-06

**Authors:** Haya A. Alfassam, Majed S. Nassar, Manal M. Almusaynid, Bashayer A. Khalifah, Abdullah S. Alshahrani, Fahad A. Almughem, Abdullah A. Alshehri, Majed O. Alawad, Salam Massadeh, Manal Alaamery, Ibrahim M. Aldeailej, Aref A. Alamri, Abdulwahab Z. Binjomah, Essam A. Tawfik

**Affiliations:** 1KACST-BWH Centre of Excellence for Biomedicine, Joint Centers of Excellence Program, King Abdulaziz City for Science and Technology (KACST), Riyadh 11442, Saudi Arabia; halfassam@kacst.edu.sa (H.A.A.); massadehsa@ngha.med.sa (S.M.); alaameryma@ngha.med.sa (M.A.); 2National Center for Biotechnology, Life Science and Environment Research Institute, King Abdulaziz City for Science and Technology (KACST), Riyadh 11442, Saudi Arabia; mnassar@kacst.edu.sa; 3National Center for Pharmaceutical Technology, Life Science and Environment Research Institute, King Abdulaziz City for Science and Technology (KACST), Riyadh 11442, Saudi Arabia; malmusaynid@kacst.edu.sa (M.M.A.); bkhalifah@kacst.edu.sa (B.A.K.); falmughem@kacst.edu.sa (F.A.A.); abdualshehri@kacst.edu.sa (A.A.A.); 4Microbiology Department, Riyadh Regional Laboratory, Ministry of Health, Riyadh 12746, Saudi Arabia; aalsharani@moh.gov.sa (A.S.A.); ialdeailej@moh.gov.sa (I.M.A.); aralalamri@moh.gov.sa (A.A.A.); 5Material Science Research Institute, King Abdulaziz City for Science and Technology (KACST), Riyadh 11442, Saudi Arabia; moalawad@kacst.edu.sa; 6Developmental Medicine Department, King Abdullah International Medical Research Center, King Saud Bin Abdulaziz University for Health Sciences, King Abdulaziz Medical City, Ministry of National Guard Health Affairs, Riyadh 11481, Saudi Arabia; 7College of Medicine, Alfaisal University, Riyadh 11533, Saudi Arabia

**Keywords:** gold nanoparticles, sialic acid, hemagglutinin, surface plasmonic resonance, viral detection, colorimetric tool, SARS-CoV-2, influenza B, MERS

## Abstract

Sialic acid that presents on the surface of lung epithelial cells is considered as one of the main binding targets for many respiratory viruses, including influenza and the current coronavirus (SARS-CoV-2) through the viral surface protein hemagglutinin. Gold nanoparticles (Au NPs) are extensively used in the diagnostic field owing to a phenomenon known as ‘surface plasmonic resonance’ in which the scattered light is absorbed by these NPs and can be detected via UV-Vis spectrophotometry. Consequently, sialic acid conjugated Au NPs (SA-Au NPs) were utilized for their plasmonic effect against SARS-CoV-2, influenza B virus, and Middle-East respiratory syndrome-related coronavirus (MERS) in patients’ swab samples. The SA-Au NPs system was prepared by a one-pot synthesis method, through which the NPs solution color changed from pale yellow to dark red wine color, indicting its successful preparation. In addition, the SA-Au NPs had an average particle size of 30 ± 1 nm, negative zeta potential (−30 ± 0.3 mV), and a UV absorbance of 525 nm. These NPs have proven their ability to change the color of the NPs solutions and patients’ swabs that contain SARS-CoV-2, influenza B, and MERS viruses, suggesting a rapid and straightforward detection tool that would reduce the spread of these viral infections and accelerate the therapeutic intervention.

## 1. Introduction

Throughout decades, viruses such as the influenza virus and coronavirus were the main culprits for several respiratory syndrome outbreaks worldwide [1]. In December 2019, a substantial worldwide pandemic emerged, which later on was named coronavirus disease 2019 (COVID-19), and has been recognized by a third Severe Acute Respiratory Syndrome strain of the *Coronaviridae* family known as SARS-CoV-2 [2]. This disease is highly contagious, affecting the lower respiratory tract, gastrointestinal tract, and central nervous systems, and it might damage some vital organs, including the heart, kidneys, and liver [3]. The first cases were detected in Wuhan, China, with patients developing pneumonia and flu-like symptoms that were raised rapidly, leading to a severe global pandemic and threatening public health [4].

Initial studies showed that the SARS-CoV-2 virus originated from animal reservoirs and was then transmitted to humans, which has been spread increasingly through aerosol droplets from an infected person to others [5,6]. Currently, there is no official report of an antiviral agent that can efficiently eradicate the virus. Hence, self-isolation, social distancing, and quarantine have been useful mandatory practices to reduce disease transmission.

Scientists have been studying coronavirus to understand more about the pathogenicity of the SARS-CoV-2 virus. The current form of such a virus is composed of a genomic RNA and a phosphorylated nucleocapsid (N) protein within phospholipid bilayers [7]. Two types of spike proteins, spike (S) glycoprotein and hemagglutinin (HA), are located on the viral-surface. Other proteins such as membrane (M) protein and envelope (E) protein make up the viral envelope [8]. It was revealed that the S protein is the leading viral fusion protein that assures the entrance and attachment of the virus to the host primary receptors [9]. Once inside the human body, the virus can hijack the host system and uses the host’s own protease enzyme to process its S protein, hence, mediate binding to a receptor located in the human lungs with the help of an enzyme known as Angiotensin-converting enzyme 2 (ACE2), leading to the infection [10].

As COVID-19 became a serious emerging epidemiological threat, an efficient clinical diagnostic tool has become an urgent need to reduce the contagiousness of the disease, to understand the virus better, and to be able to detect or even inhibit the infection in its early stages, hence, obtain a better prognosis and therapeutic outcomes. Several diagnostic tools have been used to detect different types of viruses, including the revers-transcriptase polymerase chain reaction (RT-PCR), which is used to detect the presence of viral genetic material from the isolated patient samples [11]; serological assay such as enzyme-linked immunosorbent assay (ELISA), which depends on the antibodies (i.e., IgM and IgG) produced naturally by the human defense system and found in the blood or saliva samples and which can be used as detection for the virus [12]; and Loop-Mediated isothermal amplification (LAMP) assay [13]. However, these diagnostic tests have many limitations and drawbacks such as time-consuming, high cost, low sensitivity, and the need for specialists.

Recently, gold nanoparticles (Au NPs) have gained massive attention in the field of nanomedicine due to their unique advantageous properties such as sensitivity, less toxicity, and optical detection properties [14]. The modification of Au NPs surface with specific human expressed macromolecules (i.e., glycans) can provide promising applications; for instance, targeting, probing, and biosensing [15]. Sialic acid (SA) is considered the main glycoprotein that presents on the surface of lung epithelial cells and incorporates in respiratory infection diseases caused by viruses [16]. It can be used as a binding site for the viral surface protein HA in SARS-CoV-2 and other viral types, such as influenza and Middle-East respiratory syndrome (MERS) viruses [17]. Previous studies have demonstrated the ability of the SA to bind with several types of viruses, including influenza virus, human parainfluenza virus, and coronavirus [18,19]. It was also reported that SA could be attached to spike glycoprotein that occurs on the surface of the MERS coronavirus [20].

Therefore, this study aims to develop a functionalized Au NPs conjugated with SA (SA-Au NPs) as a potential diagnostic tool for respiratory viruses, particularly SARS-CoV-2 along with influenza B and MERS viruses, to reduce the spread of these viruses and to accelerate the therapeutic intervention once an individual gets infected.

## 2. Materials and Methods

### 2.1. Materials

N-acetylneuraminic acid was purchased from Biosynth-Carbosynth (Compton, UK). Chloroauric acid solution (HAuCl_4_) was obtained from Sigma-Aldrich (Poole, UK). Sodium hydroxide (NaOH) was supplied from Loba Chemie (Mumbai, India). Deionized water was generated through Milli Q Millipore (Billerica, MA, USA) and used throughout the study.

### 2.2. Preparation of Sialic Acid-Gold NPs (SA-Au NPs)

The preparation of SA-Au NPs formulations was performed by a modified method of [18]. Briefly, approximately 7.7 mg of sialic acid (2.5 mM) was dissolved in 10 mL deionized water and mixed with 250 µL of HAuCl_4_ (0.02 M) followed by the addition of 100 µL NaOH (1.0 M), to adjusted the pH to 10 that was measured by Starter 3100 pH and conductivity bench instrument (OHAUS, Parsippany, NJ, USA). The mixture was then stirred at approximately 1000 RPM at 95 °C for 5 min. The solution color was changed from pale yellow to dark red wine color. After the solution being cooled at room temperature, the SA-Au NPs were filtered using 0.22 µm cellulose acetate membrane (falcon easy flow filter^®^) and was stored at 4 °C for further analysis.

### 2.3. UV/VIS Spectroscopy Analysis

The SA-Au NPs samples were analyzed using UV/VIS/NIR Spectrometer, Lambda 950 instrument (Perkin Elmer, Waltham, MA, USA) at a wavelength range of 200 to 850 nm. The absorbance measurements were performed using 1-cm path length quartz cuvettes. Deionized water was used as a reference. A peak of the SA-Au NPs is visible in between 524 to 531 nm representing the surface plasmon resonance (SPR) phenomena [21]. All UV analyses were performed in triplicates.

### 2.4. Particle Size and Zeta Potential Measurements

The mean diameter and zeta potential determination of SA-Au NPs were measured by the Zetasizer Nano Series (Malvern Instruments, Great Malvern, UK). The measurements were carried out at room temperature. Values reported are the mean of three measurements ± standard deviation (SD).

### 2.5. Fourier-Transform Infrared Spectroscopy (FTIR)

The conjugation of the SA and Au NPs was confirmed by Spectrum GX FT-IR System (Perkin Elmer, Waltham, MA, USA). The measurements were carried out at room temperature with a resolution of 8 cm^−1^ and an Optical Path Difference (OPD) velocity of 0.2 cm/s over a spectrum of 4000 to 400 cm^−1^. For the SA powder, a small amount (≈1 mg) was mixed with the KBr powder (IR grade) by agate mortar and pestle for 5–10 min, and then the powder mixture was placed into the sample holder for IR analysis. For HAuCl_4_ (diluted in deionized water) and SA-Au NPs solution samples, few drops of the solutions were placed on the sodium chloride (NaCl) plate. Then another free plate was placed on the top of that plate to spread the liquid drops on the surface to create a thin layer. These two plates were clamped together and mounted onto the sample holder for IR analysis. The obtained spectra were analyzed and plotted by OriginPro 2016 (OriginLab Corporation, Northampton, MA, USA). All FTIR analyses were performed in triplicates.

### 2.6. Scanning Electron Microscopy (SEM)

SA-Au NPs shape and diameter were characterized using a JSM-IT500HR SEM (JEOL, Tokyo, Japan). The NPs were measured at an accelerating voltage of 30 kV, imaged without coating. The suspended SA-Au NPs were sonicated for approximately 10 s, and then a drop of the NPs solution was deposited on a carbon-coated stub and allowed to dry at room temperature. Data were collected over a selected surface area of samples using resolution of 500 nm.

### 2.7. Transmission Electron Microscopy (TEM)

The morphology of SA-Au NPs was also assessed using a JEM-1400 TEM (JEOL, Tokyo, Japan). One drop of the NPs solution was placed onto a 400-mesh carbon-coated copper grid, then the sample was air-dried at room temperature and analyzed using an acceleration voltage of 120 kV.

### 2.8. Viral Genetic Material Confirmation Test

An ethical approval number H1RI-15-Nov20-01 was obtained from King Saud Medical City (KSMC) institutional review board (IRB), Riyadh, Saudi Arabia, for using patients’ nasopharyngeal swab samples in this study. The nasopharyngeal swabs (sampling flocked swabs) were collected from patients who were positive for influenza B, MERS-CoV, and SARS-CoV-2. All patients’ swabs were preserved in Transport Medium-2 viral transport medium (Vircell, Granada, Spain).

The Influenza B viral presence was confirmed using Xpert^®^ Xpress Flu/RSV kit on GeneXpert^®^ XVI all in one system (GeneXpert, Cepheid, Sunnyvale, CA, USA). Both MERS and SARS-CoV-2 viruses were detected using RealStar^®^ MERS-CoV RT-PCR kit 1.0 (altona Diagnostics GmbH, Hamburg, Germany) and DiaPlexQ™ Novel Coronavirus Detection kit (2019-nCoV; SolGent Co., Ltd., Daejeon, Korea), respectively, on an RT-PCR LightCycler^®^ 480 Instrument II system (Roche Molecular Systems Inc., Pleasanton, CA, USA).

### 2.9. Colorimetric Detection of Viral Particles

After confirming the presence of the viruses on patients’ swabs, the colorimetric test was performed in a modified biosafety level 3 (BSL-3) laboratory. Nasopharyngeal swabs that contain influenza B virus (20 swabs), MERS-CoV (10 swabs), and SARS-CoV-2 (10 swabs) were directly immersed in micro-centrifuge tubes containing 0.5 mL SA-Au NPs solution. Ten blank swabs (i.e., viral free swabs) were used as control for this study. After incubating the swabs in the SA-Au NPs solution for 20 min (after being optimized previously), the change of the color on the swabs and the NPs solutions were visually inspected and pictured using a camera.

### 2.10. Statistical Analysis

The mean and SD were calculated using OriginPro 2016 software (OriginLab Corporation, Northampton, MA, USA).

## 3. Results and Discussion

### 3.1. Preparation of Sialic Acid-Gold NPs (SA-Au NPs)

SA-Au NPs were prepared in a one-pot reaction, without any chemical alteration of the SA, using a modified method of [18]. In this study, HAuCl_4_ was used as a gold precursor, where SA was used as a reducing and stabilizing agent for HAuCl_4_, enabling the fabrication of Au NPs and reducing the aggregation of these NPs that might develop upon storage [22]. The SA amount was doubled, and the pH was adjusted to 10 to ensure a proper reduction and formation of Au NPs, as suggested by [23], compared to [18].

In addition, the mixture of HAuCl_4_, SA, and NaOH was heated up and stirred at 95 °C and 1000 RPM, respectively, for 5 min to allow the formation of NPs with an average particle size of 30 nm, which was the ideal size in this study, as it will be explained later. After the NPs solution was cooled down, they were purified by filtration using 0.22 µm cellulose acetate membrane instead of ultracentrifugation that was suggested by [18]. This method of purification was necessary to remove the aggregated SA-Au NPs and the non-formed components, as well as, to preserve the dark red wine color, as the color started to fade (turned into pinkish-like color) upon each centrifuge cycle. The SA-Au NPs solution was stable after this preparation method, in which it preserved its dark red wine color for at least two months sitting on the top bench and in the refrigerator. This finding was also in agreement with [18]. However, a more applicable stability study of such NPs is required as a future perspective.

### 3.2. UV/VIS Spectroscopy Analysis

The complete preparation of SA-Au NPs was confirmed visually by the color change (i.e., from yellow to dark red wine color) as shown in Figure 1. The change of color during the synthesis of SA-Au NPs was the consequence of the redshift of the plasmonic peak, which relies on the size and shape of the yielded NPs and the interparticle distance [24]. The UV-Vis spectroscopy measurement indicated the maximum SPR band absorbance position at 525 nm, as illustrated in Figure 2.

### 3.3. Particle Size and Zeta Potential Measurements

The physicochemical characterization of SA-Au NPs was performed in terms of particle size and zeta potential measurements using Zetasizer. The prepared SA-Au NPs was confirmed to be monodispersed, with an average hydrodynamic size of 30.37 ± 1.11 nm and a low polydispersity index (PDI) of 0.095. The zeta potential measurement showed that the SA-Au NPs have an anionic surface charge (−29.83 ± 0.3 mV). This particle size of approximately 30 nm was desired to maintain the dark red wine color of the NPs solution, as it will change to light red or purple color upon decreasing and increasing the particle size, respectively [24,25]. It is essential to preserve this color and size of the SA-Au NPs due to the Plasmon shift that will result from binding to viral particles allowing the size to increase, and, thus, the color turns to a more dark-reddish (or purplish) color, which could be detected visually.

### 3.4. Fourier-Transform Infrared Spectroscopy (FTIR)

To verify the structural integrity and characterize the intermolecular conformational changes of the SA-Au NPs, FTIR spectroscopy of SA, HAuCl_4_ (diluted in deionized water) SA-Au NPs was performed, as illustrated in Figure 3. Due to the five hydroxyl groups and the secondary amine group that forms the SA chemical structure, they can reduce HAuCl_4_ to synthesize Au NPs. The SA FTIR spectrum exhibited vibrational modes at 1724 and 1655 cm^−1^, representing the C=O in carboxyl and amide groups, respectively, while the stretching vibration at 1434 cm^−1^ means the carboxylic acid O-H bending. There were also multiple small peaks presented at the fingerprint region (1500 to 500 cm^−1^) indicated for substitutions in the oxane ring. This FTIR spectrum is consistent with [26,27].

The FTIR peaks for HAuCl_4_ that was diluted in deionized water showed a robust and broad peak between 3000 and 3700 cm^−1^ due to the H-bond stretching of the O-H group of the water, which also appeared in the SA-Au NPs (Figure 3). However, the SA-Au NPs FTIR spectrum showed an enhanced absorption peak at approximately 2695 cm^−1^ (C-H stretching), the C=O stretching of the SA (≈1650 cm^−1^) was maintained, while all peaks at the fingerprint region that presented in SA were disappeared. These suggest the successful conjugation of SA to the molecular chains of the Au NPs, as reported in [18].

### 3.5. Scanning Electron Microscopy (SEM) and Transmission Electron Microscopy (TEM)

The SA-Au NPs morphology was further investigated using SEM and TEM (Figure 4 and Figure 5, respectively). Both microscopies exhibited the successful synthesis of the SA-Au NPs that were equitably monodisperse spheres. This finding further justified the dark red wine color obtained upon the synthesis of the SA-Au NPs, as the shape of the Au NPs could affect the color intensity [28].

### 3.6. Colorimetric Detection of Viral Particles

The most well-known fact that respiratory viruses, such as influenza B, MERS, and SARS-CoV-2 viruses, can modify various host specificities depending on their HA protein that presents on the viral surface. These viruses can bind to the lung epithelial cells’ SA receptors via HA, which is considered one of their main binding targets [16,17,18,19,20,29]. Therefore, the SA-Au NPs were tested against influenza B, MERS, and SARS-CoV-2 viruses by directly immersing the viral-loaded swabs’ tips, which were collected from patients, into the SA-Au NPs solution. Viral-free swabs (i.e., blank swabs) were used as experiment control. After 20 min of incubation with the swabs, an apparent color change in the NPs solution was visually detected in all viral strains compared to the blank swabs that showed a similar color intensity as the free SA-Au NPs, as shown in Figure 6. These SA-Au NPs demonstrated a target-specific aggregation with the viral particles through the HA-SA binding, which displayed a Plasmon shift to a darker-red color.

In addition, the carboxylic group found in the SA affects the binding of the HA on the viral particles [18]. When the viral particle is immersed in the SA-Au NPs solution, each NP will attach to the virus surface protein HA, resulting in aggregation of these NPs on the virus surface, allowing the change in the absorption spectra toward the red shifting upon shortening the inter-particular distance of the SA-Au NPs. This will allow the color to change to more dark red or purple color, hence, signaling the presence of viral particles. Another observation was that the tip on each viral-loaded swab changed from red to yellowish-white (Figure 6). This might be due to the transfer of the viral particles from the swab’s tip (i.e., the part of the swab exposed to the nasopharyngeal region of patients) to the SA-Au NPs solution that would require a further investigation.

To confirm the binding between the NPs and the viruses, the absorbance of the influenza sample was observed by the UV-Vis spectrophotometry compared to the blank swab sample, which were performed in triplicate. There was a slight broadness in the Au NPs distinctive absorption peak (525 nm), a broadness at about 277 nm, and a formation of a more intense peak at 222 nm compared to 214 nm of the blank swab, as demonstrated in Figure 7. This could indicate a protein-like shape of the influenza virion that would need further investigation. It was previously reported that the UV absorption spectrum for the influenza virus is between 250 and 300 nm, which corresponded to the viral proteins rather than the nucleic acid (i.e., RNA) [30]. Generally, it is known that nucleic acids (DNA or RNA) have an absorption wavelength at 260 nm, while it is 280 nm for proteins [31]. Therefore, these observations by the UV-Vis spectrophotometry of the influenza sample may suggest the successful attachment of the SA-Au NPs with the viral particles, which is promising. However, more confirmatory tests are required, such as analyzing the particle size and structural integrity of the viral-bounded NPs and observing the NPs surface modification by electron microscopies.

## 4. Conclusions and Future Directions

Based on SA’s binding ability that exists on the surface of lungs epithelial cells to certain viruses, including SARS-CoV-2, it is possible to develop SA stabilized Au NP system that can be used as a detection tool through its plasmonic shift upon binding to viruses. These NPs are able to change color in the presence of viral surface protein HA. Therefore, a rapid and straightforward one-pot synthesis method was utilized to prepare SA-Au NPs more efficiently than the previously reported synthesis methods, which required multiple synthesis and purification steps. These NPs were successfully designed with a perfect dark red wine color, an average particle size of 30 ± 1 nm, and a UV absorbance of 525 nm representing the SPR of Au NPs in general. SA was used as a reducing and stabilizing agent for these Au NPs, in which SEM and TEM confirmed the spherical shape of such NPs, proving their successful fabrication and the results of the FTIR that suggested the integration of SA within the NPs. Upon evaluating the SA-Au NPs against three respiratory viral strains; SARS-CoV-2, influenza B, and MERS (all known to bind to SA), these NPs were able to change the color of the NPs solutions to darker red (or even purple) color due to the presence of the viruses, as well as, the color of patients’ nasopharyngeal swabs. These preliminary results might indicate the potential use of this NP system as a diagnostic tool that could help with early therapeutic intervention and reduce the spread of viral infections through their viral-binding ability. More work is currently on-going to specifically target SARS-CoV-2, particularly by conjugating the SA-Au NPs with SARS-CoV-2 specific antibodies to be used as a potential theranostic tool for COVID-19.

## Figures and Tables

**Figure 1 pharmaceutics-13-00502-f001:**
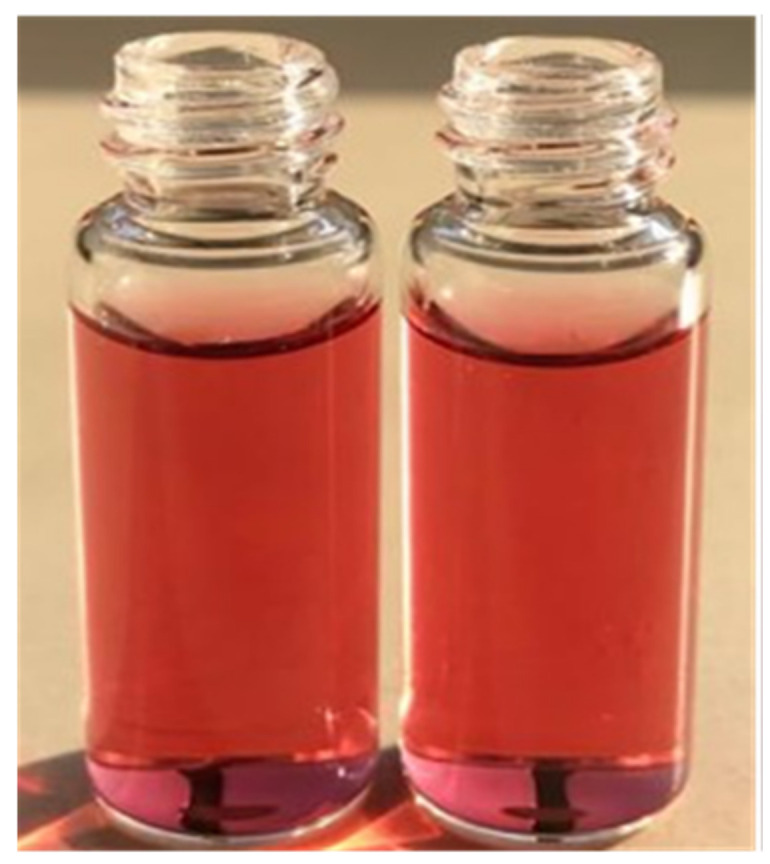
A representative image of the prepared nanoparticles (NPs), in which the dark red wine color indicated the successful preparation of sialic acid conjugated Au NPs (SA-Au NPs).

**Figure 2 pharmaceutics-13-00502-f002:**
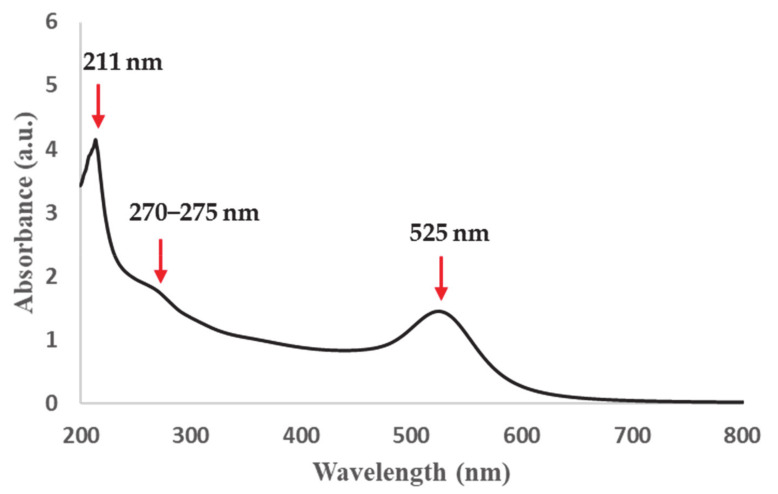
UV–Vis absorbance spectra of SA-Au NPs showing the SPR band at 525 nm.

**Figure 3 pharmaceutics-13-00502-f003:**
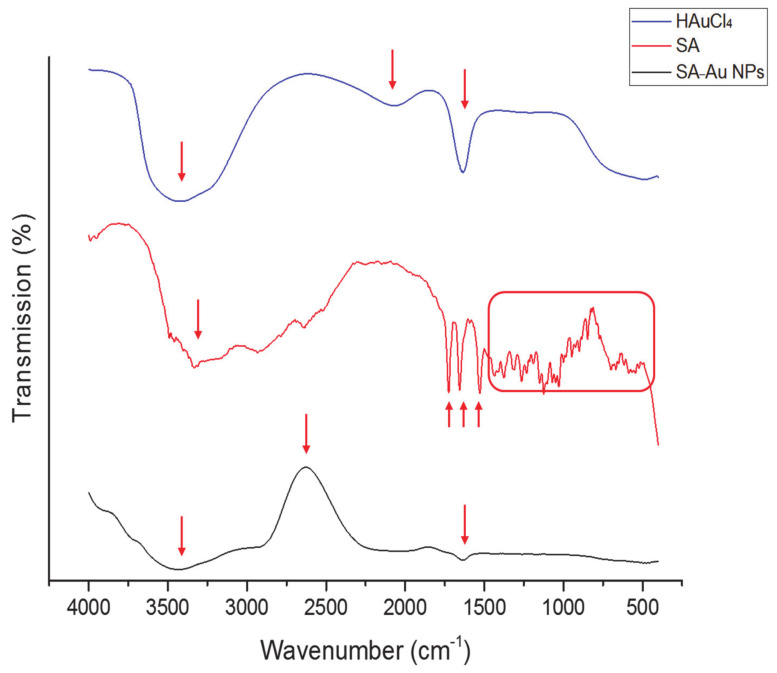
FTIR spectroscopy transmissions of SA, HAuCl_4_ (diluted in deionized water), and SA-Au NPs, showing the distinctive peaks of SA at 1724, 1655, 1434 and between 1500 to 500 cm^−1^ that disappeared in the SA-Au NPs, while an enhanced absorption peak at approximately 2695 cm^−1^ and the C=O stretching at 1650 cm^−1^ that presented in these NPs spectrum suggesting the successful preparation of the NPs.

**Figure 4 pharmaceutics-13-00502-f004:**
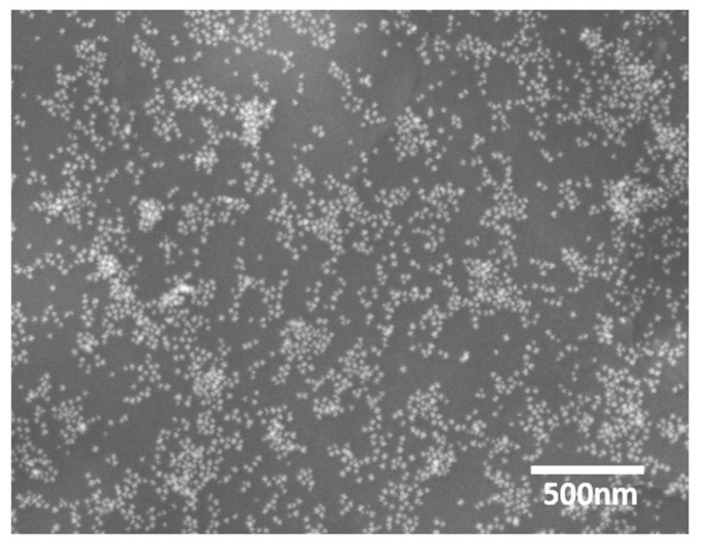
SEM images of SA-Au NPs showing the sphere-like shape of the prepared NPs measure at a resolution of 500 nm.

**Figure 5 pharmaceutics-13-00502-f005:**
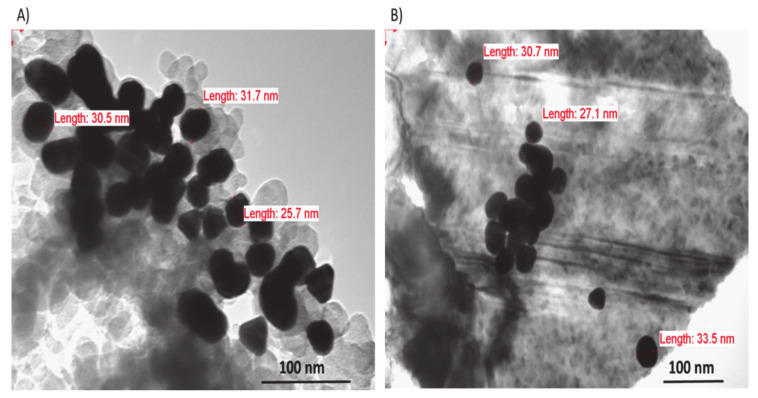
TEM image of SA-Au NPs showing the spherical shape of prepared with different individual particle sizes. (**A**) Measurement at a magnification of 300,000×; (**B**) Measurement at a magnification of 200,000×.

**Figure 6 pharmaceutics-13-00502-f006:**
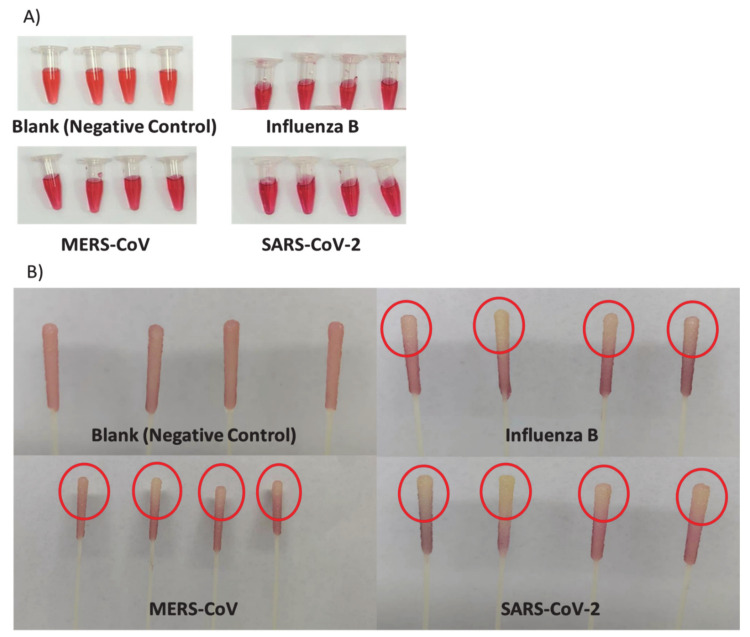
Positive swab samples of influenza B, Middle-East respiratory syndrome (MERS), and SARS-CoV-2 were immersed in SA-Au NPs for 20 min compared to the blank swabs. (**A**) An evident color change in the NPs solution in all viral-loaded swabs compared to the blank swabs; (**B**) Yellowish-white tips of all viral-loaded swabs compared to the tips of the blank swabs, which shown in red.

**Figure 7 pharmaceutics-13-00502-f007:**
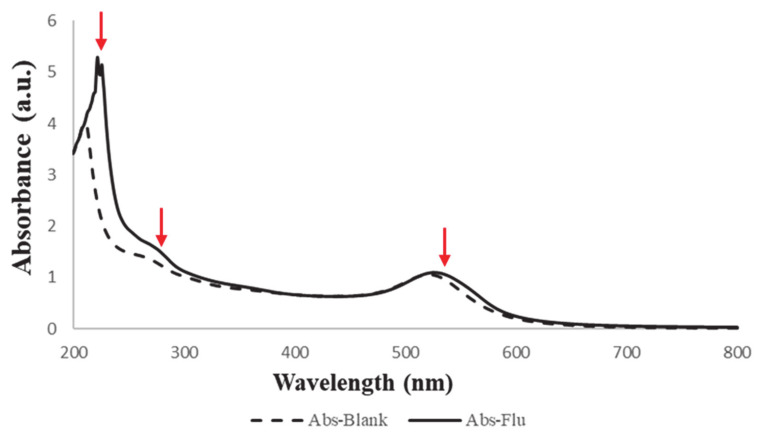
UV–Vis absorbance spectra of SA-Au NPs after been immersed with a positive influenza swab (Abs-Flu) compared to a negative blank swab (Abs-Blank) that show a slight broadness of the SPR band at 525 nm, a broadness at ≈277 nm and the formation of an intense peak at 222 nm that was originally at 214 nm.

## Data Availability

Not applicable.

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
