# Peer review of "Development of a Colorimetric Tool for SARS-CoV-2 and Other Respiratory Viruses Detection Using Sialic Acid Fabricated Gold Nanoparticles"

_pharmaceutics, 2021, doi:10.3390/pharmaceutics13040502_

Round 1

Reviewer 1 Report

  1. Figure 2 - Placing markers for individual values would improve the appearance of the graph.
    Was the UV-Vis analysis done by the authors in Section 3.2 performed once? - The authors did not include information on the number of replicates, and this is important in assessing the reliability of the results.
  2. In Section 3.6, the authors show color brightening as a result of contact with the virus suspension. Unfortunately, there is no information on how the control was prepared for which no such color change was observed - this is important so that it can be confirmed that the color change is related to the attachment of the viruses and not other components in the suspension - I the authors comment.
  3. Figure 7 - no information on whether the authors performed only a single experimental trial - please complete this information.
  4. In the absence of an unambiguous determination of what factor causes color changes in NP. suspensions, it is difficult to assume that the method presented by the authors is a potential diagnostic method.

Author Response

Please find our response attached

Reviewer 2 Report

The manuscript “Development of a Colorimetric Tool for SARS-CoV-2 and Other Respiratory Viruses Detection Using Sialic Acid Fabricated Gold Nanoparticles”. It is a very interesting work describing the use of SA-AuNPs for the detection of SARS-CoV-2, 35 influenza B, and MERS viruses, and could be reduce the spread of these viral infections and accelerate the therapeutic intervention. I recommend this manuscript to be published in Pharmaceutics with a few recommendations.

Line 179. The authors need to show that the SA does not suffer any chemical alteration after to obtain de AuNPs.

Line 224. “HAuCl4” should be “HAuCl4

Line 234. “SA-AU NPs” should be “SA-Au NPs”

Figure 3. “HAuCl4” should be “HAuCl4

Figure 4. Why the resolution of the SEM images is different? 

All the references should be in the same format 

Author Response

Please find our response attached

Round 2

Reviewer 1 Report

Accept in present form